# Peer review of "TanSat ACGS on-orbit spectral calibration by use of individual solar lines and entire atmospheric spectra"

_Atmospheric Measurement Techniques, 2020_

## Referee Comment (RC1) · Anonymous Referee #1 · 6 Jul 2020

The manuscript deals with spectral calibration of the ACGS sensor on TanSat. Two spectral calibration methods are explained and insight is given on the spectral variations and stability of the sensor. The information on the choices made on the calibration approach and the spectral stability of the sensor is of interest to the greenhouse gas retrieval community. Overall, the topic is considered adequate for publication in AMT. A revision of the manuscript is needed before publication addressing the issues identified below.

General Comments

1. Abstract and introduction fall short in explaining the role of the two methods and

to which kind of observation they are applied. Later in the manuscript it seems that Method 1 (using individual Fraunhofer lines) is applied to solar irradiance measurements only and allows the determinations of ground-to-orbit shifts and to monitor day to day variability as well as longer-term trends, and that Method 2 (using entire atmospheric spectra) is applied to Earth radiance measurements only and allows also the determination of intra-orbital variations. Please clarify this upfront, already in the abstract and the introduction.

2. It is stated that Method 2 is used for verifying Method 1 (Section 3.2) and that close consistency has been found (abstract). Please clarify how this verification has been made and what exactly has been compared for this purpose. Have orbital averages from Method 2 been take for this? A figure with results from both methods depicted over the orbital phase would help to illustrate the level of consistency. Have longer time series of the two methods been compared? Please add such figures.

3. It is not clear whether the two spectral calibration methods are employed only for off-line analysis by hand or to which degree they are implemented in the systematic processing op to Level-1 (as suggested at Line 110 for Method 1). Please clarify this upfront.

4. Please introduce the spectral requirement(s). Are there separate requirements on a) shifts with respect to the ground characterisation, b) on in-flight spectral stability (at Level-0 prior to spectral calibration), and c) in-flight spectral knowledge at Level-1b (after spectral calibration). Please introduce the requirements explicitly. It would be nice if the spectral calibration requirements were introduced with a short discussion on the tracing to the $CO_2$ product uncertainty requirement.

5. The L1b users are certainly interested in the performance of the two methods, which is unfortunately not reported. If possible, please give estimates for the uncertainty of the spectral axis obtained by the two methods.

Specific Comments

Abstract line 9: it is stated that the observed spectral variations are partly caused by vibrations. This is not understood. It is expected that mechanical vibrations within the spectrometer, in particular causing displacement of the slit with respect to the detector can cause a widening of the instrument spectral response, but not a shift of its barycentre.

Please introduce a blank space between numbers and units throughout the manuscript.

Line 39: When discussing the aim of the study ("only study the wavelength offset or shift with respect to pre-launch spectral calibration.") please distinguish between ground-to-orbit change and in-orbit variability. Also please include a discussion of the spectral knowledge needs.

Line 40: It is argued that the ILS can be assumed to be constant on orbit "because a common diffuser is used for solar observation". This is not understood. In what sense and across which elements is the diffuser "common"? Across the spectral bands? It is not clear how diffuser features or the use of different diffusers would possibly introduce spectral variation.

Line 41: It is referred to "decon events". Please expand / clarify what is meant: de-contamination"?

Line 41: Please complete the discussion of "decon events" by stating whether such events could affect spectral variability, e.g. by changes in thermo-mechanical loads within the spectrometer.

Line 46: Please change "methods to evaluate the ACGS's wavelength calibration connecting each focal plane array (FPA) pixel to a specific wavelength." to "methods to assign a specific wavelength to each focal plane array (FPA) pixel of the ACGS."

Line 55: expand "shift and squeeze" to "shift and squeeze of the spectral axis".

Line 61: Typo: change "telescope aperture" to "spectrometer aperture". Section 2.1 It is not explained how the coefficients of the spectral axis (Eq 1) are derived from the

in-flight spectral shifts determined at individual Fraunhofer lines. Please clarify.

Line 71: please add: the number of SPECTRAL pixels in the . . . bands.

Line 72: please use the label "O2A band" as introduced earlier for the ACGS spectral band throughout the manuscript (as opposed to the "O2 A-band" as labelled by Fraunhofer in 1814).

Line 72: typo: change "s" to "a"

Line 72: please change "two pixels per full width at half maximum (FWHM)" to "two pixels per spectral resolution increment (defined as the full width at half maximum (FWHM) of the ILS"

Line 76: change "in three band" to "in the three bands"

Line 77: what is meant with "some middle pixels"? in the centre of the field? In the spectral centre of the bands?

Line 84: Please justify the adequacy of this parameterization e.g. by discussing the expected or observed smoothness of the wavelength as a function of pixel number. The polynomial used is a function of the index p rather than the index difference with respect to the centre pixel index. This asymmetry would cause instabilities to occur near the band edge with higher index numbers, if they occur. Please discuss if this is relevant here.

Line 87: Please clarify what is meant with "the sampling resolution" of the Kurucz's spectra. Is it spectral sampling or spectral resolution?

Line 87: The solar reference spectrum by Chance and Kurucz (JQSRT, 2010) has a spectral resolution of 0.04 nm and a spectral sampling of 0.01 nm. Please correct or clarify. Please clarify which spectrum is used for which spectral domain, as the solar reference spectrum by Chance and Kurucz does not cover wavelengths larger than 1000 nm. Please specify spectral sampling and resolution of the reference spectrum

by Fontela et al.

Line 90: The statement that the spectral resolution of the solar reference spectra is one order of magnitude higher than the spectral resolution of the ACGS seems not valid (see comment on Line 87). Please correct or clarify.

Line 109: change "are" to "is"

Line 114: Reformulate "The wavelength offsets of each band have annually variation in a year." Proposed "The wavelength offsets of each band exhibit an annual variation."

Line 116: Reformulate "There are little thermal gradient" to something like "thermal gradients are small", please quantify.

Line 117: The slit is missing in the listing of relevant optical component in this context. Please add.

Line 118: changeS

Line 118: It is stated that "offset which are similar for each FOV". Please clarify if that is expected because of the spectrometer design or whether this is simply a finding of this analysis.

Line 124: Please add "insensitivity" to "insensitivity of the spectral response"

Line 126: it is not clear in which sense the detector material is relevant for the sensitivity to thermal variations. Please explain. Is it about thermal expansion of the detector?

Line 129: Change "in UV-visible bands" to "in the O2A band". ACGS does not cover the UV.

Equation 3: The contributions to the cost function are weighted by the inverse of the measurement noise, so the weight is lower in the Fraunhofer and their wings lines as compared to the continuum. It is not clear whether this weighting strategy is useful, in view of the fact that the spectral information is exactly in these spectral regions. Please

discuss / consider.

Line 142: It is not satisfactory to state that "the search routine" is used to find the minimum of equation (3). Please specify the minimization routine, at least by specifying its class / type, maybe the library from which this routine is taken.

Line 148: The statement "Hence, the result of this method is independent of solar lines and can be compared with the calibration results shown in section 2." Is confusing. Please discuss in which sense the two methods bring different information and what has been learnt. Please discuss the parts of the orbit in which solar irradiance and Earth radiance spectra are acquired and in which the two methods are applied. Please discuss the thermal behavior of the spectrometer as a function of the orbital phase.

Line 155: the statement "Totally, ACGS has 1242x9, 500x9 and 500x9 different ILS tables in O2 A-band, WCO2 and SCO2 band, respectively." Is not understood. Are so many ILSs stored in the tables?

Line 170: Please clarify which Earth radiance scenarios are eligible for the spectral calibration. It is assumed that the O2A band all scenes are eligible, while in the SCO2 and WCO2 bands dark ocean scenes might have to be excluded due to low signal and hence low signal to noise ratio levels. It is expected that cloudy scenes are eligible for spectral calibration in view of the high signal to noise ratio and the presence of Earth atmospheric signatures.

Line 176: Please discuss the implication of the observation that "shifts derived from this method agree closely with that calculated from solar spectra". Please discuss what can be learnt on the orbital variations.

Figures 4-6 caption: Please specify for which year(s) the data are plotted. Clarify whether the wavelength changes shown are averages of spectral shifts determined at a set of Fraunhofer lines, or band-averaged shifts.

Figure 6 caption: Typo. Change "spectral resolution in WCO2 band" to "spectral resolution in SCO2 band".

Figures 7-9 caption: Please specify over which domain and range the statistics are evaluated (is it the temporal variation in the domain as shown in Figures 4-6?

Figure 10 caption: Please mention that results from the second method applied to Earth radiance spectra are shown. Add labels to the three panels indicating the spectral bands.

Figure 11 caption: Change "plus squeezed" to "and squeezed"

Table 2 Caption. It is not clear over what exactly the mean and the standard deviation are evaluated. Is it the statistics shown in Figures 7-9? Is it the spectral variation in all irradiance spectra or in all radiance spectra acquired on the specified day (or something else).

---

## Short Comment (SC1) · 18 Jul 2020

Many thanks to the anonymous referee. These comments are very valuable and have greatly improved the quality of the manuscript. We have revised the manuscript following the comments.

**General Comments**

1. Abstract and introduction fall short in explaining the role of the two methods and to which kind of observation they are applied. Later in the manuscript it seems that Method 1 (using individual Fraunhofer lines) is applied to solar irradiance measurements only and allows the determinations of ground-to-orbit shifts and to monitor day to day variability as well as longer-term trends, and that Method 2 (using entire atmospheric spectra) is applied to Earth radiance measurements only and allows also the determination of intra-orbital variations. Please clarify this upfront, already in the abstract and the introduction.

Reply: We accept this very valuable comment and have revised the abstract and the introduction, where the role of the two methods and to which kind of observation they are applied are clarified.

2. It is stated that Method 2 is used for verifying Method 1 (Section 3.2) and that close consistency has been found (abstract). Please clarify how this verification has been made and what exactly has been compared for this purpose. Have orbital averages from Method 2 been take for this? A figure with results from both methods depicted over the orbital phase would help to illustrate the level of consistency. Have longer time series of the two methods been compared? Please add such figures.

Reply: This comment and comment 1 make us further consider the role and relationship of the two approaches. As reference standards, the Fraunhofer lines possess much higher stability in nature than the synthetic spectra calculated by radiative transfer model (RTM). Therefore, the second method cannot be used to verify the first method. But, the second method can obtain the intra-orbit variation to complement the first one. We revised the 'verifying' to the 'complement' in abstract and section 3.2. In fact, for the assessment of the performance of ACGS, the first method is used as the main spectral calibration method because of its high accuracy and stability.

A simple comparison over Beijing in 2017 is performed for the two methods. We select these orbits according to some conditions. Only the cloud-free and clean scenes are selected because of the little aerosol effects. Dark ocean scenes are also excluded. Only 7 orbits are selected. We calculate the shifts in these orbits using the second method. Then the shifts in one orbit are averaged to obtain a shift. These averaged

AMTD
shifts are compared with those derived from the first method. For all field of view, the mean differences in O2A, WCO2 and SCO2 bands are -0.0017 nm, 0.0016 nm and 0.0020 nm, respectively. We add the figure of comparison. The intra-orbit shifts from the second method on 27 September 2017, is also provided as another case that is similar to that case on 23 April 2017 in Figure 10.

3. It is not clear whether the two spectral calibration methods are employed only for off-line analysis by hand or to which degree they are implemented in the systematic processing op to Level-1 (as suggested at Line 110 for Method 1). Please clarify this upfront.

Reply: The first method (using individual Fraunhofer lines) is used to process all the solar measurements to monitor the spectral variations on orbit. After the solar measurements are available, this method will automatically process the solar data. But the second method (using entire atmospheric spectra) is employed only by hand to some orbits. It is more complicated than the first method because it depends on the atmospheric condition (cloud and aerosol), the accuracy of RTM and the reanalysis data. Sometimes, the results are not outputted because the iterations fail in convergence.

4. Please introduce the spectral requirement(s). Are there separate requirements on a) shifts with respect to the ground characterisation, b) on in-flight spectral stability (at Level-0 prior to spectral calibration), and c) in-flight spectral knowledge at Level-1b(after spectral calibration). Please introduce the requirements explicitly. It would be nice if the spectral calibration requirements were introduced with a short discussion on the tracing to the CO2 product uncertainty requirement.

Reply: The spectral requirements include band coverage, spectral resolution, spectral sampling in FWHM, and the spectral calibration accuracy. Dr. Zhongdong Yang shows them in detail in the paper 'Laboratory spectral calibration of the TanSat atmospheric carbon dioxide grating spectrometer' published in 2018. Please refer to that paper about the detailed spectral requirements.
For TanSat, there are no separate spectral requirements at stage of ground, Level-0 and Level-1b. The requirement on in-flight spectral accuracy is 5-10% of the spectral resolution, which has been added to Table 1 and also clarified at the end of the third paragraph in Section 1. The aim of this manuscript is to analyze if this spectral calibration requirement is satisfied. These requirements on the tracing to the CO2 product uncertainty requirement exceed the field of this manuscript, because CO2 product uncertainty depends on too many factors, such as your retrieval method(physical or regression), radiometric calibration at Level-1, instrumental model(SNR, ILS ...), and so on.

5. The L1b users are certainly interested in the performance of the two methods, which is unfortunately not reported. If possible, please give estimates for the uncertainty of the spectral axis obtained by the two methods.

Reply: The first method has the high accuracy and stability in nature based on the Fraunhofer lines. We show its performance in Figure 4-6 and Figure 7-9. Figure 4-6 show that the first method has the ability to reveal the variation trend. Sorry for that we did not explain the error bar in figure 7-9. The horizontal axis is the spectral axis, the standard deviation is shown by the error bar for each FOV and each Fraunhofer line in the year 2017. After averaged for all FOVs and all Fraunhofer lines, the standard deviations in O2A, WCO2 and SCO2 bands are about 0.0039 nm, 0.013 nm and 0.021 nm, respectively. These estimations have been added at the last paragraph in Section 2.2.

The second method has the ability to reveal the spectral variation too. This method is not developed by ourselves. We simply apply the second method to our work without a full performance assessments made by ourselves. We strongly recommend the L1b users to refer to the paper published by Jos H.G.M. van Geffen and Roeland F. van Oss (Applied Optics, 2003), which is also given in the references. They develop this method to estimate spectral variations of GOME by using the earthshine spectra. Also, they shows the performance of this method.
**Specific Comments**

Abstract line 9: it is stated that the observed spectral variations are partly caused by vibrations. This is not understood. It is expected that mechanical vibrations within the spectrometer, in particular causing displacement of the slit with respect to the detector can cause a widening of the instrument spectral response, but not a shift of its barycentre.

Reply: We accept this comment and revise the statement. 'Vibrations' is deleted from the sentence at line 9.

Please introduce a blank space between numbers and units throughout the manuscript.

Reply: We have added a blank space throughout the manuscript.

Line 39: When discussing the aim of the study ("only study the wavelength offset or shift with respect to pre-launch spectral calibration.") please distinguish between ground-to-orbit change and in-orbit variability. Also please include a discussion of the spectral knowledge needs.

Reply: We accept this valuable comments and have revised the statement. '... study the wavelength offset or shift with respect to pre-launch spectral calibration' is changed to '... study the wavelength ground-to-orbit change with respect to pre-launch spectral calibration and the wavelength in-orbit variability.'

And, we have added the statement that shows the spectral knowledge needs at the end of the this paragraph.

Line 40: It is argued that the ILS can be assumed to be constant on orbit "because a common diffuser is used for solar observation". This is not understood. In what sense and across which elements is the diffuser "common"? Across the spectral bands? It is not clear how diffuser features or the use of different diffusers would possibly introduce spectral variation.

AMTD
Reply: The transmissive diffuser is a special diffuser that allows light to transmit to reduce solar irradiance. This diffuser can subtly change the ILS due to its uneven illumination of the telescope aperture (see Sun et al., 2017). TanSat's diffuser is a 'general' or 'common' diffuser that only allows light to be reflected. We think this diffuser will not change the ILS.

Line 41: It is referred to "decon events". Please expand / clarify what is meant: decontamination"? Line 41: Please complete the discussion of "decon events" by stating whether such events could affect spectral variability, e.g. by changes in thermomechanical loads within the spectrometer.

Reply: We have learned 'decon events' from the paper of Sun et al. (AMT, 2017). For OCO-2, it is necessary to decontaminate the ice(decon events) that accumulates on the antireflective (AR) coating of the FPAs in O2 A band. 'This effect will enhanced the reflectance of the FPA, and the reflected light might be scattered back to the FPA by the other optical components. This effect can be quantified as widening of the ILS wings'.

For TanSat, there are not this decon events. Therefore, the on-orbit spectral calibration of TanSat is easier than that of OCO-2.

Line 46: Please change "methods to evaluate the ACGS's wavelength calibration connecting each focal plane array (FPA) pixel to a specific wavelength." to "methods to assign a specific wavelength to each focal plane array (FPA) pixel of the ACGS."

Reply: We have changed this sentence.

Line 55: expand "shift and squeeze" to "shift and squeeze of the spectral axis".

Reply: We have expanded this sentence.

Typo: change "telescope aperture" to "spectrometer aperture". Section 2.1 It is not explained how the coefficients of the spectral axis (Eq 1) are derived from the in-flight spectral shifts determined at individual Fraunhofer lines. Please clarify.

AMTD
Reply: The 'telescope aperture' has been changed to 'spectrometer aperture'. These coefficients of the spectral axis (Eq 1) have been derived from the spectral calibration test on ground. After launch, the coefficients C2-C5 remain unchanged to describe the non-linear effects of the wavelength on pixel index. In the first method, the C1 also remains unchanged. The C0 coefficients for each FOV and each band are corrected by subtracting the shifts. At each position of the selected Fraunhofer lines for each FOV and each band, a individual shift will be calculated at the reference position by: 'shift = measurement - reference'. Then, the shifts for a FOV in a band will be averaged to get a shift.

We have added above statements at the end of Section 2.1.

Line 71: please add: the number of SPECTRAL pixels in the ... bands.

Reply: Spectral is added.

Line 72: please use the label "O2A band" as introduced earlier for the ACGS spectral band throughout the manuscript (as opposed to the "O2 A-band" as labelled by Fraunhofer in 1814).

Reply: The label 'O2A band' is used throughout the manuscript.

Line 72: typo: change "s" to "a". Line 72: please change "two pixels per full width at half maximum (FWHM)" to "two pixels per spectral resolution increment (defined as the full width at half maximum (FWHM) of the ILS"

Reply: 's' has been changed to 'a' at line 72. "two pixels per full width at half maximum (FWHM)" is changed to "two pixels per spectral resolution increment (defined as the full width at half maximum (FWHM) of the ILS".

Line 76: change "in three band" to "in the three bands"

Reply: 'in three band' is changed to 'in the three bands' throughout the manuscript.

Line 77: what is meant with "some middle pixels"? in the centre of the field? In the
**spectral centre of the bands?**

Reply: 'some middle pixels' is very confused. We have revised this to 'three adjacent pixels located in the central section of the FPA for the three bands.'

Line 84: Please justify the adequacy of this parameterization e.g. by discussing the expected or observed smoothness of the wavelength as a function of pixel number. The polynomial used is a function of the index p rather than the index difference with respect to the centre pixel index. This asymmetry would cause instabilities to occur near the band edge with higher index numbers, if they occur. Please discuss if this is relevant here.

Reply: Thanks for this valuable comment. This parameterization that uses the index of pixels had been identified in the lab testing on ground in 2014. We did not notice this asymmetry effect. Another instrument called Greenhouse gas Absorption Spectrometer (GAS), the design of which is similar to the ACGS, is being developed for FenYun-3 satellite series. We will compare the two types of parameterizations.

Line 87: Please clarify what is meant with "the sampling resolution" of the Kurucz's spectra. Is it spectral sampling or spectral resolution?

Reply: Sorry for the confused expression. We have checked the Kurucz's spectra. "the sampling resolution of 0.001nm" should be 0.001 nm spacing. We have revised this at line 87.

Line 87: The solar reference spectrum by Chance and Kurucz (JQSRT, 2010) has a spectral resolution of 0.04 nm and a spectral sampling of 0.01 nm. Please correct or clarify. Please clarify which spectrum is used for which spectral domain, as the solar reference spectrum by Chance and Kurucz does not cover wavelengths larger than 1000 nm. Please specify spectral sampling and resolution of the reference spectrum by Fontela et al.

Reply: Kurucz's spectrum atlas are widely used for radiative transform model and data
retrieval in the scientific community. We have found the two paper published by Fontela, Chance and Kurucz when we look for the solar spectrum. Because it introduces the reference spectrum in many details from 200-1000nm, we list them as the references in our manuscript. Indeed, the paper shows the solar spectra with 0.04nm resolution and 0.01 nm sampling. And, this spectrum was developed for GOME, SCIAMACHY et al.

But the resolution of the spectrum used in our research are not the same as that described by Chance and Kurucz's paper. The spectrum in O2A band with spacing of 0.001 nm are also released via his web(http://kurucz.harvard.edu/sun/IRRADIANCE2005/). Also, the high resolution of near 0.001 nm in WCO2 and SCO2 band can be found in another directory of IRRADIANCE2005.

I guess these spectra used in my research might be produced or updated after Chance and Kurucz's paper was published, because in the header of spectra, Kurucz writes that "this work was financially supported in part by the NIES GOSAT Project, Center for Global Environmental Research National Institute for Environmental Studies, Japan". So, the bands expand from O2A to CO2 bands. At current, the resolution of Kurucz's solar spectra is the highest among our all solar reference spectra. Some solar spectra provided by the RTM model can not meet our needs.

Line 90: The statement that the spectral resolution of the solar reference spectra is one order of magnitude higher than the spectral resolution of the ACGS seems not valid (see comment on Line 87). Please correct or clarify.

Reply: We have deleted this statement at line 90.

Line 109: change "are" to "is"

Reply: 'are' is changed to 'is' at line 109.

Line 114: Reformulate "The wavelength offsets of each band have annually variation
in a year." Proposed "The wavelength offsets of each band exhibit an annual variation."

Reply: This sentence at line 114 has been reformulated.

Line 116: Reformulate "There are little thermal gradient" to something like "thermal gradients are small", please quantify.

Reply: This sentence is changed to 'The small thermal gradients can lead to the slight change of the geometry of major optical components such as slit, collimator, grating and FPAs.'

Line 117: The slit is missing in the listing of relevant optical component in this context. Please add.

Reply: slit is added.

Line 118: changeS

Reply: 'change' is revised to 'changes'.

Line 118: It is stated that "offset which are similar for each FOV". Please clarify if that is expected because of the spectrometer design or whether this is simply a finding of this analysis.

Reply: That is expected. We have revised this statement to 'offset which is expected to be similar for each FOV'.

Line 124: Please add "insensitivity" to "insensitivity of the spectral response".

Reply: 'insensitivity of the spectral response' has been added at line 124.

Line 126: it is not clear in which sense the detector material is relevant for the sensitivity to thermal variations. Please explain. Is it about thermal expansion of the detector?

Reply: The original statement explaining the effect of temperature on larger shifts is inaccurate. The sentence 'The O2 A-band uses the silicon .... to minor variations in temperature' has been deleted. From the perspective of the whole ACGS design, O2A
band locates in the middle of the instrument, while the two CO2 bands locates on the both sides. The thermal gradients in CO2 bands are larger than that in O2A band.

Line 129: Change "in UV-visible bands" to "in the O2A band". ACGS does not cover the UV.

Reply: Yes, we have changed this sentence.

Equation 3: The contributions to the cost function are weighted by the inverse of the measurement noise, so the weight is lower in the Fraunhofer and their wings lines as compared to the continuum. It is not clear whether this weighting strategy is useful, in view of the fact that the spectral information is exactly in these spectral regions. Please discuss / consider

Reply: We did not consider this question when we use this method. We simply apply this equation developed by Geffen et al. (2003) in our work. In principal, this weighting strategy is reasonable because the measurement contains noise which can affect the convergence of the cost function. Re-examining this weighting strategy demands clear physical insight about the two kinds of spectra, i.e, measurement and simulation. We think this need to be further studied.

Line 142: It is not satisfactory to state that "the search routine" is used to find the minimum of equation (3). Please specify the minimization routine, at least by specifying its class / type, maybe the library from which this routine is taken.

Reply: We have checked our routine to identify that fmincon routine in MATLAB package is used to find the minimum of a nonlinear function.

Line 148: The statement "Hence, the result of this method is independent of solar lines and can be compared with the calibration results shown in section 2." Is confusing. Please discuss in which sense the two methods bring different information and what has been learnt. Please discuss the parts of the orbit in which solar irradiance and Earth radiance spectra are acquired and in which the two methods are applied. Please
**discuss the thermal behavior of the spectrometer as a function of the orbital phase.**

Reply: We want to express that the two methods use different references and different processing techniques to obtain the spectral variations. Then, the results of two methods can be compared. The statement 'Hence, the result of this method is independent of solar lines and can be compared with the calibration results shown in section 2' is revised to 'The result of this method can be compared with that shown in section 2'.

In the reply to comment 2, we discuss the parts of the orbit in which the results from solar irradiance and Earth radiance spectra are compared. These discussions have been added to the third paragraph in Section 3.2.

The author and co-authors have few knowledge about the thermal behavior of the ACGS as a function of the orbital phase. We had a talk with the expert who knows the thermal variation, and learned that the thermal behavior of ACGS is very complex because it can be affected by different observation modes which are nadir, glint, target and solar modes. The temperature is cooled to -30 degree Celsius for FPAs in CO2 bands and to -5 degree Celsius for other components. In addition, The ACGS and CAPI are integrated designed. The platform controls the thermal balance for the two instruments. The temperature maybe has fluctuations of  $\pm 2$  degree in one orbit.

Line 155: the statement "Totally, ACGS has 1242x9, 500x9 and 500x9 different ILS tables in O2 A-band, WCO2 and SCO2 band, respectively." Is not understood. Are so many ILSs stored in the tables?

Reply: Yes, the all ILSs are stored in the Level-1b files in HDF5 format.

Line 170: Please clarify which Earth radiance scenarios are eligible for the spectral calibration. It is assumed that the O2A band all scenes are eligible, while in the SCO2 and WCO2 bands dark ocean scenes might have to be excluded due to low signal and hence low signal to noise ratio levels. It is expected that cloudy scenes are eligible for spectral calibration in view of the high signal to noise ratio and the presence of Earth
**atmospheric signatures.**

Reply: We clarify the conditions for calibration in the reply to comment 2. Yes, Dark ocean scenes are excluded. Cloudy scenes are also excluded because we can not simulate the measurement due to the limitation of RTM when clouds exist.

Line 176: Please discuss the implication of the observation that "shifts derived from this method agree closely with that calculated from solar spectra". Please discuss what can be learnt on the orbital variations.

Reply: We think that the agreements imply that the spectral performance of ACGS spectrometer is stable on orbit. The abnormal variations determined from solar spectra can provide warning signature for the instrument status. The second method provides the intra-orbit variation and the first method provides longer-term variation trends. Therefore, they complement each other.

Figures 4-6 caption: Please specify for which year(s) the data are plotted. Clarify whether the wavelength changes shown are averages of spectral shifts determined at a set of Fraunhofer lines, or band-averaged shifts.

Reply: The year is 2017. The wavelength changes are averages at the selected Fraunhofer lines. We have added the explanation in the text where the figure is cited.

Figure 6 caption: Typo. Change "spectral resolution in WCO2 band" to "spectral resolution in SCO2 band".

Reply: Yes, this figure is for SCO2 band.

Figures 7-9 caption: Please specify over which domain and range the statistics are evaluated (is it the temporal variation in the domain as shown in Figures 4-6?

Reply: The statistics are evaluated based on each individual Fraunhofer line in each band in 2017. The positions of each lines are also shown in Figure 3. The bars represent the standard deviations. We have revised the caption to 'The static-
stics of wavelength change derived from individual solar line for all spatial FOVs in O2A/WCO2/SCO2 band in 2017' Figure 4-6 show the averaged change in a band for the all selected Fraunhofer lines shown in Figure 3.

Figure 10 caption: Please mention that results from the second method applied to Earth radiance spectra are shown. Add labels to the three panels indicating the spectral bands.

Reply: The caption is revised to 'The wavelength change derived from the second method applied to Earth radiance spectra. These results are for each FOV and each band along latitude in one orbit on April 23, 2017 (DOY 113). The top panel is for the O2A band, the middle panel is for the WCO2 band and the bottom panel is for the SCO2 band.'

Figure 11 caption: Change "plus squeezed" to "and squeezed"

Reply: 'plus' has been changed to 'and'.

Table 2 Caption. It is not clear over what exactly the mean and the standard deviation are evaluated. Is it the statistics shown in Figures 7-9? Is it the spectral variation in all irradiance spectra or in all radiance spectra acquired on the specified day (or something else).

Reply: It is the statistics shown in Figure 10. It is the spectral variation in all radiance spectra in one orbit on April 23, 2017.
**Fig. 1.** Comparison of shifts form the two methods for nine FOVs in WCO2 for the orbits over Beijing, China, in 2017. The shifts derived from the second method are averaged in one orbit for each FOV.

---

## Referee Comment (RC2) · Anonymous Referee #3 · 1 Sep 2020

Review of Bi et al., "TanSat ACGS on-orbit spectral calibration by use of individual solar lines and entire atmospheric spectra"

In this work, the authors discuss using individual solar lines and fitting to entire atmospheric spectra of TanSat ACGS data to fit for shifts and stretches in the wavelength (dispersion) calibration of the ACGS relative to its preflight calibration.

There are a number of problems with this paper. Space-borne spectrometers require calibration in terms of radiometric, geometric (geolocation), spectral, and polarization behavior. In terms of spectral calibration, both wavelength (dispersion) calibration as well as calibrating the instrument line shape (ILS) functions. Of all these categories of calibration, fitting the dispersion is probably the easiest and best understood. It is so straightforward and robust, in fact, that few groups even bother to do this in a dedicated fashion, because it is easily fit for simultaneously with other atmospheric parameters required to derive e.g. column mean carbon dioxide concentration (XCO2) or solar-induced chlorophyll fluorescence (SIF). This is well documented in many publications (e.g., Reuter et al. 2010, Taylor et al 2011, Frankenberg 2011, O'Dell et al 2012, Crisp et al. 2017, Wu et al. 2018).

Therefore, the methods espoused are nothing new. It is confusing to me why they focus on individual solar lines, when they could just as easily fit for example a 2 or 3-parameter update to the wavelength dispersion by using all the solar lines (or all solar lines greater than a certain depth); this would be significantly more robust than using a single solar line. And also why they choose the Kurucz spectrum as their reference, where nearly all groups have found that the Toon solar reference spectrum (http://mark4sun.jpl.nasa.gov/toon/solar/solar_spectrum.html) is considered superior in terms of accuracy.

But these are minor problems. The biggest problem is that much of this work is already documented in a prior publication by many of these same authors, in the recent paper "Inflight Performance of the TanSat Atmospheric Carbon Dioxide Grating Spectrometer" (Yang et al., 2020). That publication also fits for spectral shifts using the solar spectrum, and produces time series of the results, just as in this paper. Remarkably, the Bi et al. paper under review here does not even reference, discuss, or compare to the results from Yang et al. in terms of the wavelength shifts found in both works. The single reference to the Yang et al. paper in this work is to state the size of the spatial field-of-view of TanSat; nothing about its results on spectral shifts relative to preflight.

Another serious problem with this paper is that although they say that the solar and atmospheric methods give similar results, they never actually demonstrate this. There is no figure that compares them, no method to actually discuss and compare quantitatively their respective results. I tried to do this manually, and they actually did not agree (comparing the offsets given in their table 2 results, to those in figures 4-6 for the solar method).

A minor point is that while they discuss the solar doppler shift (induced by the relative motion of the spacecraft and the sun, when they use the solar method), they never discuss the doppler

shift of the telluric (earth's atmosphere) lines, induced by the relative motion of the satellite and the target point on the earth's surface.  It's not clear if they take this into account.

Finally, there are many English grammar errors that require correcting.  I do not bother to list them, as this is a relatively minor point considering the structural deficiencies in the paper.

Therefore, I recommend to reject this work for publication.  It introduces nothing new to the field, its results are not coherent, it does not try to explain the physics of what is going on (e.g. the noticeable jump in spectral shift around DOY 150 in Figure 4).  It does not explain the disparity of results across the 9 FOVs, in particular in the strong $CO_2$ band.   So it is not new, it does not explain anything to us regarding what is happening with TanSat in particular, the results are confusing and not well-presented, and the effect is easily and automatically taken into account by the TanSat $XCO_2$ and SIF retrievals anyway (e.g., Liu et al., 2013, Du et al., 2018).

**References**

Crisp, David, Harold R. Pollock, Robert Rosenberg, Lars Chapsky, Richard AM Lee, Fabiano A. Oyafuso, Christian Frankenberg et al. "The on-orbit performance of the Orbiting Carbon Observatory-2 (OCO-2) instrument and its radiometrically calibrated products." *Atmospheric Measurement Techniques* 10, no. 1 (2017): 59-81.

Du, Shanshan, Liangyun Liu, Xinjie Liu, Xiao Zhang, Xingying Zhang, Yanmeng Bi, and Lianchong Zhang. "Retrieval of global terrestrial solar-induced chlorophyll fluorescence from TanSat satellite." *Science Bulletin* 63, no. 22 (2018): 1502-1512.

Frankenberg, C., A. Butz, and G. C. Toon. "Disentangling chlorophyll fluorescence from atmospheric scattering effects in O2 A-band spectra of reflected sun-light." *Geophysical Research Letters* 38, no. 3 (2011).

Liu, Yi, DongXu Yang, and ZhaoNan Cai. "A retrieval algorithm for TanSat XCO 2 observation: Retrieval experiments using GOSAT data." *Chinese Science Bulletin* 58, no. 13 (2013): 1520-1523.

O'Dell, C. W., B. Connor, H. Bösch, D. O'Brien, C. Frankenberg, R. Castano, M. Christi et al. "The ACOS CO 2 retrieval algorithm–Part 1: Description and validation against synthetic observations." *Atmospheric Measurement Techniques* 5, no. 1 (2012): 99-121.

Reuter, M., M. Buchwitz, O. Schneising, J. Heymann, H. Bovensmann, and J. P. Burrows. "A method for improved SCIAMACHY CO 2 retrieval in the presence of optically thin clouds." *Atmospheric Measurement Techniques* 3, no. 1 (2010): 209-232.

Taylor, Thomas E., Christopher W. O'Dell, Denis M. O'Brien, Nobuyuki Kikuchi, Tatsuya Yokota, Takashi Y. Nakajima, Haruma Ishida, Dave Crisp, and Teruyuki Nakajima. "Comparison of cloud-screening methods applied to GOSAT near-infrared spectra." *IEEE Transactions on Geoscience and Remote Sensing* 50, no. 1 (2011): 295-309.

Wu, Lianghai, Otto Hasekamp, Haili Hu, Jochen Landgraf, Andre Butz, Ilse Aben, David F. Pollard et al. "Carbon dioxide retrieval from OCO-2 satellite observations using the RemoTeC algorithm and validation with TCCON measurements." (2018): 3111.

---

## Author Comment (AC2) · 11 Sep 2020

article setspace color

[Figure]

**Reply to interactive comments**

September 11, 2020

**1 Reply to interactive comments**

Thanks for the comments, however, most of these comments are incorrect, and show the reviewer's lack of knowledge of on-orbit wavelength calibration. These comments are very likely to mislead the readers into believing that wavelength calibration on orbit is an easy thing.

1. There are a number of problems with this paper. Space-borne spectrometers require calibration in terms of radiometric, geometric (geolocation), spectral, and polarization behavior. In terms of spectral calibration, both wavelength (dispersion) calibration as well as calibrating the instrument line shape (ILS) functions. Of all these categories of calibration, fitting the dispersion is probably the easiest and best understood. It is so straightforward and robust, in fact, that few groups even bother to do this in a dedicated fashion, because it is easily fit for simultaneously with other atmospheric parameters required to derive e.g. column mean carbon dioxide concentration (XCO2) or solar-induced chlorophyll fluorescence (SIF). This is well documented in many publications (e.g., Reuter et al. 2010, Taylor et al 2011, Frankenberg 2011, O'Dell et al 2012, Crisp et al. 2017, Wu et al. 2018).

Reply: What are a number of problems? We have not found them in this paragraph.

How do you know few groups even bother to do this? Groups are what kinds of groups? calibration group, retrieval group or application group? If you search spectral calibration or wavelength calibration, you will find many papers focusing on this topic.

Fitting the dispersion between two spectra is not so easy, so straightforward and robust. On the contrary, it is very complex because of the fitting algorithm, the possible multi-solutions depending on the degree of convergence. Especially, when other atmospheric parameters are derived with the dispersion based on optimal estimation, for instance, the XCO2 or SIF, fitting between the observation and the simulated atmospheric spectra will become more complex because the dispersion and other parameters are competitive to contribute to the goodness of fit. If the fitting failed, no reasonable results are outputted. The failure often occurs in the retrieval. If the offset is off by more than about one spectral sample, the fit will often fail to converge. So, the fit is not always robust, easy and straightforward.

All these papers given by the reviewer are not for spectral calibration, but for data retrieval. They just mention the correction of the residual spectra shifts in the process of XCO2 retrieval. For example, Reuter et al. (2010) mentions the wavelength shift in one sentence in section 3.2.1. Taylor et al (2011) writes "...a wavelength multiplier (f). The wavelength multiplier is necessary as the wavelength calibration given in the TANSO-FTS L1B files does not account for either short-term drifts in the scan laser frequency ...", They use a multiplier to correct the wavelength bias rather than dispersions because GOSAT uses an interferometer, not a grating spectrometer. The reviewer confuses the basic calibration terminology between different instruments, which may mislead readers and editors. Because these papers focus on retrieval, rather than spectral calibration, we do not think the spectral calibration is well documented in the given papers. Therefore, these papers are not listed in our manuscript.

The author and co-authors are the members of the TanSat calibration team and our

responsibility is providing well calibrated spectral radiance data for the users who use the spectra data for XCO2 retrieval. So, we have to calibrate the satellite observations. We believe that the user communities are interested in this spectral calibration which is also considered in their retrieval.

2. Therefore, the methods espoused are nothing new. It is confusing to me why they focus on individual solar lines, when they could just as easily fit for example a 2 or 3-parameter update to the wavelength dispersion by using all the solar lines (or all solar lines greater than a certain depth); this would be significantly more robust than using a single solar line. And also why they choose the Kurucz spectrum as their reference, where nearly all groups have found that the Toon solar reference spectrum (http://mark4sun.jpl.nasa.gov/toon/solar/solar_spectrum.html) is considered superior in terms of accuracy.

Reply: No paper shows the same method as that shown in section 2 in our manuscript. So, the methods using individual Fraunhofer lines is new. The method described in section 3 is developed by Geffen and van Oss (2003). In principle, direct computation of spectral shift based purely on individual Fraunhofer lines position is more robust than matching two entire spectra, as explained above. Second, according to our experience, the computation of using individual solar lines is faster than that of fitting two entire spectra because the iterations are avoided.

Different solar spectra have different accuracies. The accuracy of intensity has little effect on the spectral calibration because what is needed is the position of Frauhofer lines. This is different to XCO2 retrieval where the accuracy of intensity has large effects on the XCO2. The absolute intensity can be removed by normalization or scaling the spectra in the wavelength calibration, as noted in section 2.D by van Geffen JH (2003). It is not important which solar spectra is used in this paper. More important thing is how to perform the spectral calibration on orbit to assess the wavelength variation. Therefore, Kurucz spectrum can be used as reference. This spectrum can well satisfy TanSat calibration requirements. Also, this spectrum is widely used in science

community. Only this spectrum is available when we began to study this method in the year 2014.

3. But these are minor problems. The biggest problem is that much of this work is already documented in a prior publication by many of these same authors, in the recent paper "Inflight Performance of the TanSat Atmospheric Carbon Dioxide Grating Spectrometer" (Yang et al., 2020). That publication also fits for spectral shifts using the solar spectrum, and produces time series of the results, just as in this paper. Remarkably, the Bi et al. paper under review here does not even reference, discuss, or compare to the results from Yang et al. in terms of the wavelength shifts found in both works. The single reference to the Yang et al. paper in this work is to state the size of the spatial field-of-view of TanSat; nothing about its results on spectral shifts relative to preflight.

Reply: This comment confuses the two types of problems. It is well known to all that the biggest problem should be big errors in principle, big defects in method, or big errors in conclusion. Reference documents are not part of the big problems.

"The single reference..." is not right. We do reference professor Yang's two papers. Please see the last two papers in the references in the manuscript. When this manuscript was prepared, the latest Yang's paper was just accepted, and had not been published. So we did not know how to cite an unpublished paper.

If you compare professor Yang's paper and this manuscript, you will find that much of this work is not documented in the professor Yang's paper which describes the infight performance of the TanSat ACGS. He summaries the inflight spectroscopic performance in section III.A, as well as ILS, radiometry, dark current, SNR, gain coefficient and so on. This manuscript describes the spectral calibration method using individual solar line in many details and compares this with the fitting method using entire atmospheric spectra. Therefore, much of this work is not documented in Yang's paper.

The reviewer do not really read Yang's paper. This comment, "That publication also fits for spectral shifts using the solar spectrum ...,", is not right. Please note that professor

Yang also uses the available Fraunhofer lines method to get the wavelength shift, but not fitting the solar spectra to derive the spectral shift. He describes this method in the second paragraph in section III.A. Now, Dr. Yang's paper has been published, we cite this paper in section 1.

4. Another serious problem with this paper is that although they say that the solar and atmospheric methods give similar results, they never actually demonstrate this. There is no figure that compares them, no method to actually discuss and compare quantitatively their respective results. I tried to do this manually, and they actually did not agree (comparing the offsets given in their table 2 results, to those in figures 4-6 for the solar method).

Reply: A comparison over Beijing in 2017 is performed for the two methods (see the new figure 1 in this document). We select these orbits according to certain conditions. Only the cloud-free and clean scenes are selected because of the little aerosol effects. Dark ocean scenes are also excluded. A total of 7 orbits are selected. We calculate the shifts for these orbits using the second method. Then the shifts in one orbit are averaged to obtain a shift. These averaged shifts are compared with those derived from the first method. For all field of view, the average differences in O2A, WCO2 and SCO2 bands are -0.0017 nm, 0.0016 nm and 0.0020 nm, respectively. We add the figure of the comparison. The intra-orbit shifts from the second method on 5 April 2017, is also provided as another case that is similar to that case on 23 April 2017 in Figure 10.

We think the above comparison shows that results from the two methods agree very well. And, table 2 only shows an example that the 2nd method can get the reasonable shift.

5. A minor point is that while they discuss the solar doppler shift (induced by the relative motion of the spacecraft and the sun, when they use the solar method), they never discuss the doppler shift of the telluric (earth's atmosphere) lines, induced by the

relative motion of the satellite and the target point on the earth's surface. It's not clear if they take this into account.

Reply: Yes, we correct them. The 4th co-author Dr. Liu is responsible for the geometric calibration of TanSat. The new figure 2 in this document shows the doppler shift between the instrument and the sun in solar mode in O2A band in 2017. The new figure 3 in this document shows the doppler shift variations between the instrument and the surface targets in nadir mode in O2A band in 2017. We do not show these figures in the manuscript because the doppler shift calculation is of high school students. The satellite velocity calculated by Dr. Liu is stored in the variable FrameGeometry/satellite_velocity in the L1B file that are available in the web site (http://satellite.nsmc.org.cn/portalsite/default.aspx?currentculture=en-US).

Finally, there are many English grammar errors that require correcting. I do not bother to list them, as this is a relatively minor point considering the structural deficiencies in the paper.

Reply: Several reviewers have listed them, and I have revised these English grammar errors. I also do not bother to list your syntax errors in your comments.

Therefore, I recommend to reject this work for publication. It introduces nothing new to the field, its results are not coherent, it does not try to explain the physics of what is going on (e.g. the noticeable jump in spectral shift around DOY 150 in Figure 4). It does not explain the disparity of results across the 9 FOVs, in particular in the strong CO2 band. So it is not new, it does not explain anything to us regarding what is happening with TanSat in particular, the results are confusing and not well-presented, and the effect is easily and automatically taken into account by the TanSat XCO2 and SIF retrievals anyway (e.g., Liu et al., 2013, Du et al., 2018).

Reply: The reviewer obviously do not have the relevant expert knowledge about the instrument spectral calibration and its status assessments. We have developed the individual solar method to obtain the only confirmed solvable directly. There is no doubt

that this method is new. Also, this method successfully detects the jump in spectral shift around DOY 150. We explain this jump caused by the switching of strategy of solar calibration in the 2nd paragraph in section 2.2. If you do not understand that this switching can break the raw heat balance to cause spectral shift, please you study some basic knowledge about instrument calibration.

The quite little differences among the 9 FOVs are normal. The key point is that they have coherent variation. Also, the calibration accuracies for each of the FOVs meet our requirements.

HOW VERY INTERESTING it is to say that the effect is easily and automatically taken into account!

It looks that the two papers given by the reviewer support these comments, but in fact it's not. The two papers are about products retrieval and do not demonstrate the effect is easily and automatically taken into account. In the paper by Liu et al. (2013), he only says that shifts are estimated from the spectra, but we do not know how to estimate, and this retrieval is tested using data from GOSAT which uses an interferometer, rather than grating spectrometers used by TanSat. The spectral calibration for the two types of instruments are very different. The reviewer uses the word of 'automatically'. We do not understand what this word means in retrieval? Does it mean that people should not carefully consider this shift?

In the paper by Du et al. (2018), Professor Yanmeng Bi is the fourth author. He contributes the method for radiance calibration and wavelength calibration to this paper. So, he is the co-author in Du's paper. In this work, the spectra with unreasonable wavelength shift due to the instrument performance or calibration error, were simply removed during the retrieval process. NO SHIFT WAS ESTIMATED!

**2 Summary**

Most of these comments are not conducive to improve the calibration methods, and are not conducive to enhancing the conclusions of our manuscript. We can feel that the reviewer may be familiar with the retrieval of CO2, but has very little knowledge about instrument and spectral calibration. He confuses the basic terminology between interferometer and grating calibration, and does not really understand the goals and methods for assessment of instrument status on orbit. He thinks the fit of two entire spectra is the easiest method. Actually, he does not understand the nature of fitting two spectra to get multi-variables that are competitive in the process, and can lead to improper results. Particularly, these results are not unique based on the goodness of fit. Finally, fitting often fails if one of variables becomes abnormal in the iterations. He does not understand why new method is needed for spectral calibration on orbit.

Therefore, all these comments have no real values. Most of them are very wrong, very malicious and are very likely to mislead the readers into believing that wavelength calibration on orbit is an easy thing.

But we still work hard to reply these comments point by point, and hope this reviewer can understand the nature of spectral calibration. This hope becomes one of the aims of this manuscript. Other aim is to provide useful information to the users of TanSat products.

**WCO2**

*Figure: scatter plot with x-axis "Day of Year" (50 to 400) and y-axis "shift (nm)" (-0.025 to 0.02). Legend shows red asterisks for "solar" and blue circles for "atmos."*

**Fig. 1.** The comparison of wavelength shifts form the two methods for nine FOVs in WCO2 band for the orbits over Beijing, China, in 2017.

[Figure]

**Fig. 2.** The doppler shift between the instrument and the sun in solar mode in O2A band in 2017.

**Fig. 3.** The doppler shift variations between the instrument and the surface targets in nadir mode in O2A band in 2017.

---

## Editor Comment (EC1) · Ulrich Platt (Editor) · 12 Sep 2020

Dear Dr. Yanmeng Bi,

Your manuscript was reviewed by two anonymous referees. Both reports come to the conclusion that the manuscript has serious deficiencies and can not be published in its present form or even a revised form. Particular criticism is directed to the lack of novelty of the described work and the fact that key findings (e.g. equivalence of the solar and atmospheric calibration methods) are not demonstrated in the manuscript. Also there is the problem that part of your results are already published by some of the authors of the present manuscript. I find your response to both reviewers' criticism

not convincing. In fact, instead of properly reacting to referee #3's criticism, you rather chose to attempt to draw the referee's competence into doubt. In view of the fact that two independent reviewers find severe problems with the manuscript, which are not convincingly answered by your response, I have serious doubts that the manuscript can be published and therefore I have to discourage you to submit a revised version of the manuscript.

With very best regards Ulrich Platt